# Effect of Gestational Age at Fetoscopic Laser Photocoagulation on Perinatal Outcomes for Patients with Twin–Twin Transfusion Syndrome

**DOI:** 10.3390/jcm12051900

**Published:** 2023-02-28

**Authors:** Li-Chun Chiu, Yao-Lung Chang, An-Shine Chao, Shuenn-Dyh Chang, Po-Jen Cheng, Yi-Chiao Liao

**Affiliations:** Department of Obstetrics and Gynecology, Chang Gung Memorial Hospital, Linkou Medical Center, Chang Gung University College of Medicine, 5, Fu-Shin Street, Kweishan, Taoyuan 333, Taiwan

**Keywords:** twin–twin transfusion syndrome, fetoscopic laser photocoagulation, gestational age, fetal survival, preterm premature rupture of membranes

## Abstract

Purpose: The aim of this study was to evaluate the effect of gestational age (GA) at the time of fetoscopic laser photocoagulation (FLP) for severe twin–twin transfusion syndrome (TTTS) on perinatal outcomes in a single center in Taiwan. Materials and methods: Severe TTTS was defined as a diagnosis of TTTS before a GA of 26 weeks. Consecutive cases of severe TTTS treated at our hospital with FLP between October 2005 and September 2022 were included. The evaluated perinatal outcomes were preterm premature rupture of membranes (PPROM) within 21 days of FLP, survival 28 days after delivery, GA at delivery, and neonatal brain sonographic imaging findings within 1 month of delivery. Results: We included 197 severe TTTS cases; the mean GA at the time of FLP was 20.6 weeks. After the cases were divided into cases of FLP at early (below 20 weeks) and late GAs (more than 20 weeks), the early-GA group was discovered to be associated with a deeper maximum vertical pocket in the recipient twin, a higher rate of PPROM development within 21 days of FLP, and lower rates of survival of one or both twins. In the cases of stage I TTTS, the rate of PPROM within 21 days of FLP was higher in the group that underwent FLP at an early GA than in the group that underwent FLP at a late GA (50% (3/6) vs. 0% (0/24), respectively, *p* = 0.005). Logistic regression analysis revealed that the GA at the time of FLP and the cervical length before FLP is implemented are significantly associated with the survival of one twin and the incidence of PPROM development within 21 days of FLP. The GA at the time of FLP, the cervical length before FLP, and TTTS being stage III TTTS were associated with the survival of both twins after FLP. Neonatal brain image anomalies were associated with GA at delivery. Conclusions: FLP being performed at an earlier GA is a risk factor for lower fetal survival and PPROM development within 21 days of FLP in cases of severe TTTS. Delaying FLP for cases involving stage I TTTS diagnosed at an early GA without risk factors, such as maternal symptoms, cardiac overload in the recipient twin, or a short cervical length, may be considered, but whether delaying FLP would improve surgical outcomes and, if so, how long the delay should be may need further trials to answer.

## 1. Introduction

Twin–twin transfusion syndrome (TTTS) is a complication that occurs in approximately 9% of monochorionic diamniotic twin pregnancies [1]. TTTS is caused by an imbalanced placental transfusion that leads to hypervolemia in one twin and hypovolemia in the other [2]. Fetoscopic laser photocoagulation (FLP) is the first-line therapy for TTTS diagnosed before 26 weeks of gestation [3,4,5] because FLP considerably reduces the likelihood of morbidity and mortality and achieves more favorable outcomes than does serial amnioreduction [6]. Although FLP is widely accepted and performed as a treatment for severe TTTS, it is significantly associated with a high incidence of morbidity and mortality related to preterm premature rupture of membranes (PPROM) after the operation, fetal demise, and neonatal neurological deficits after interventions [7].

Quintero staging is often used to categorize the severity of TTTS [8], and this staging system has been reported to have a high prognostic value [9,10]. However, the staging system does not account for obstetric factors, such as gestational age (GA) at diagnosis [7]. FLP is generally performed immediately when stage II to IV TTTS is diagnosed before a GA of 26 weeks. In some medical centers, FLP may not be immediately performed for stage I TTTS if the mother does not exhibit symptoms such as preterm labor or has a short cervical length [11]. However, a report indicated that initiating FLP for patients with stage I TTTS without risk factors such as a short cervical length can lead to more favorable outcomes than observation [12]. Furthermore, although several studies have reported that initiating FLP to treat TTTS at an early GA is associated with poorer perinatal outcomes [7,13], other studies have reported similar outcomes for FLP initiated at early and late GAs [14,15]. These conflicting findings indicate that the association between the GA at which FLP is initiated to treat severe TTTS and perinatal outcomes remains unclear [12,16].

A report evaluating the association between the GA at which FLP is initiated and perinatal outcomes revealed that FLP initiation at an early GA significantly and independently affects perinatal survival, PPROM occurrence after FLP, and neonatal morbidity, with poorer outcomes. The clinical trial mainly included cases of stage II to IV TTTS and only a few cases of stage I TTTS [7]. The medical center involved in the present study is the largest center in Taiwan that performs FLP for stage I to IV TTTS diagnosed before a GA of 26 weeks. The present study was conducted to evaluate the effect of the GA at which FLP is initiated and other antenatal factors, namely, the cervical length before FLP, placental location, and maximum vertical pocket (MVP) of the recipient twin, on the perinatal outcomes of patients with stage I to IV TTTS diagnosed before a GA of 26 weeks. This study was conducted in a single center and analyzed consecutive cases.

## 2. Materials and Methods

This retrospective study included patients with TTTS diagnosed before a GA of 26 weeks who were treated with FLP between October 2005 and September 2022. The TTTS diagnosis was determined on the basis of Quintero’s criteria [8]. This study was approved by the Institutional Review Board of Chang Gung Medical Hospital (IRB No. 202201763B0). Approximately 1 h and 30 min before FLP was performed, 50 mg of indomethacin was administered as a tocolytic, and 2 g of cefazolin was administered as a prophylactic antibiotic. Surgery was performed in an operating room under regional or local anesthesia, and selective laser photocoagulation of communicating vessels [17] with or without the Solomon technique was employed [18,19]. The procedure was performed using a 2 mm 0° fetoscope (Storz 26008 AA; Karl Storz GmbH, Tuttlingen, Germany) or a 3.5 mm 30° fetoscope (Storz 26008 BUA, Karl Storz GmbH, Tuttlingen, Germany) for a placenta that was mainly anterior. A working channel was placed on top, and a 0.6 mm laser fiber was inserted into the channel. The anastomoses were selectively coagulated with 10 to 30 W continuous diode laser beams on the basis of the diameter of the targeted vessel. In addition, 500 mg of cefazolin was administered once every 6 h for the 24 h following the operation, and nifedipine was employed as a tocolytic agent if necessary [20]. After FLP was completed, amnioreduction was performed to reduce the amount of amniotic fluid to the MVP of the brecipient twin less than 8 cm. The cervical length was routinely checked before FLP was initiated through at least three vaginal sonography measurements, and the value of the measurement with the shortest length was selected. Cervical cerclage was performed if the cervical length was shorter than 2 cm or funneling of the cervical canal was discovered. Cord occlusion was considered the first-line procedure for TTTS only in cases involving a major fetal anomaly.

The inclusion criteria for this study were as follows: (1) having a diagnosis of TTTS at a GA of 16 to 26 weeks and (2) having received FLP before a GA of 26 weeks at our hospital. Cases were excluded if (1) the patient received selective termination after FLP, (2) a major fetal structure or genetic anomaly was discovered after FLP, (3) the patient was lost to follow-up after FLP, or (4) the patient had a triplet pregnancy with TTTS.

Fetal survival was defined as survival 28 days after delivery. PPROM after FLP was defined as PPROM occurring within 21 days of FLP being performed. A placenta was considered to be mainly anterior when the main placental mass was located anteriorly and a 30° fetoscope was required for FLP [21]. Quintero stages were used to categorize the severity of TTTS [8], with the most severe stages considered to be Quintero stages III and IV.

All live neonates received cranial ultrasound examinations within 1 month of delivery. If more than one cranial ultrasound examination was performed, the results of the most recent examination were selected for analyses. Neonatal brain ultrasound image anomalies have been previously described in the literature [22].

Statistical analysis was conducted using SPSS (version 11.0 for Windows; SPSS, Chicago, IL, USA). The normality of the data was tested, and Student’s *t*-test or the Mann–Whitney *U* test was used to compare continuous variables between groups on the basis of these normality results. Qualitative data were compared using the χ^2^ test or Fisher’s exact test. One-way ANOVA was used to compare GA at delivery for cases of TTTS at different Quintero stages after FLP. A *p* value of less than 0.05 was considered significant. Logistic regression was used to determine the variables for fetal survival, PPROM, and neonatal brain ultrasound image anomalies after FLP. Adjusted odds ratios were obtained in forward conditional mode, where a variable was added to the mode when *p* was less than 0.05. Variables were removed when the *p* value was greater than 0.1. The odds ratios of the variables are expressed as odds ratios (95% confidence intervals).

## 3. Results

After the exclusion of two cases of selective termination due to a discordant fetal major brain image anomaly in one fetus in a twin pair, two cases of self-termination due to personal factors, one case of truncus arteriosus in the donor twin, two cases of triplets, one case of discordant meconium peritonitis with polyhydramnios in the donor twin, one case of concordant trisomy 21 being detected after FLP, one case of donor tetralogy of Fallot, and two cases that were lost to follow-up, 197 cases of patients with severe TTTS who received FLP were included (Figure 1).

The characteristics of the patients with TTTS who received FLP are listed in Table 1.

The mean GA at which FLP was implemented was 20.6 weeks. We divided patients into those in whom FLP was implemented at an early GA (≤20 weeks) and late GA (>20 weeks) and compared their characteristics and perinatal outcomes. The cases of TTTS with FLP implemented at an early GA were associated with deeper MVPs in the recipient twin, higher rates of PPROMs development within 21 days of FLP, a lower proportion of stage I TTTS, and lower survival rates of at least one or both of the twins compared with those having TTTS in whom FLP was implemented at a late GA (Table 2).

From March 2015 onward, our center began performing FLP using selective laser photocoagulation of communicating vessels with the Solomon technique in cases with suspicious anastomoses. Cases in which this photocoagulation was applied did not significantly differ from cases in which it was not applied with respect to the survival of one and two fetuses, PPROM development after FLP, and GA at delivery outcomes (Appendix A).

In cases of stage I TTTS, the risk of developing PPROM after FLP when FLP was initiated at an early and late GA was 50% (3/6) and 0% (0/24), respectively (*p* = 0.005).

We analyzed the differences in the outcomes after FLP (i.e., survival of at least one twin, survival of both twins, development of PPROMS after FLP, and GA at delivery) among individuals with different Quintero stages of TTTS (Table 3); we discovered no significant differences in the survival rate of one twin, the development of PPROMS after FLP, and GA at delivery outcomes. However, the survival rate of both twins differed between patients with different Quintero stages of TTTS. Stage III TTTS was associated with the survival of both twins after FLP. Therefore, Quintero staging was determined to influence the survival of both twins after FLP is employed.

Logistic regression was used to identify the significant antenatal variables for the occurrence of PPROM within 21 days of FLP (Table 4), the survival of both twins (Table 5), and the survival of one twin (Table 6). The GA at which FLP is implemented and the cervical length before FLP were determined to be significant factors associated with the survival of one twin and the incidence of PPROM within 21 days of FLP. The GA at which FLP is implemented, the cervical length before FLP, and stage III TTTS were determined to be associated with the survival of both twins after FLP.

After GA at delivery was incorporated as a variable for predicting neonatal brain sonographic anomalies, an earlier GA at delivery (prematurity) was discovered to be the only variable associated with the occurrence of fetal brain image anomalies (Table 7).

## 4. Discussion

In the present study, we discovered that the GA at which FLP is initiated and the cervical length before FLP were significant factors associated with fetal survival and the risk of developing PPROM after FLP in cases of severe TTTS. Prematurity was discovered to be significantly associated with neonatal brain ultrasound image anomalies.

Since FLP being initiated at an earlier GA was found to be associated with poorer outcomes, delayed FLP has been considered for cases of stage I TTTS without maternal symptoms or a diagnosis of cardiac overload in the recipient twin at an early GA [7]. A randomized trial reported that early fetal surgery is unlikely to benefit asymptomatic pregnant women with stage I TTTS and with a long cervix [11], but 60% of individuals with stage I TTTS under expectant management facing advanced staging require rapid transfer to a surgical center [4,11]. So, though delaying FLP for an early diagnosis of stage I TTTS may be beneficial to fetal outcomes, FLP can also be offered without a delay after detailed consultation with such individuals. Further research trials are required to determine the GA at which an individual with stage I TTTS without risk factors would benefit from delaying FLP and to determine the length of delay that would most improve fetal outcomes.

Even when cerclage was performed for cases involving a short cervical length or cervical funneling before FLP, the cervical length before FLP was a significant factor influencing fetal survival and the risk of developing PPROM after FLP. Although FLP and cerclage placement in individuals with TTTS with short cervical lengths were reported to be technically feasible [23], a report indicated that cerclage placement in individuals with TTTS with short cervical lengths before FLP did not increase the FLP-to-delivery interval, GA at delivery, live birth rate, or neonatal survival compared with an expectant group [24]. Because we performed cerclage placement in individuals with TTTS with short cervical lengths and a short cervical length before FLP is a risk factor for lower fetal survival, the risk of developing PPROM after FLP in individuals with such placement was higher according to our logistic regression analysis. This caused individuals with TTTS with cerclage placement after FLP in our series to have lower survival rates for one twin, a higher risk of developing PPROM after FLP, and a lower GA at delivery (Appendix A). Our findings and those of another report [24] indicated that cerclage placement in cases involving a short cervical length before FLP is implemented for severe TTTS may not significantly influence the outcomes of FLP.

FLP being implemented at an earlier GA was discovered to be a risk factor for PPROM development after FLP. Another study demonstrated that individuals who underwent FLP at a GA of earlier than 16 weeks experienced significantly higher rates of chorioamniotic separation than the standard laser group did and experienced higher rates of PPROM development after FLP and of chorioamnionitis [25]. Postoperative membrane separation was reported to be associated with an increased risk of PPROM development at ≤21 days after FLP [26]. The association between FLP being implemented at an earlier GA and the risk of PPROM development after FLP in our study may have been caused by FLP implementation at an earlier GA involving a higher risk of membrane separation.

Previously, in a study in which we recruited our first 100 individuals with TTTS who received FLP, the fetal survival rate was, respectively, highest and lowest in individuals with stage I and stage III TTTS [10]. In the present study, a low proportion of individuals in the group with FLP implemented at an early GA had stage I TTTS. This finding, in combination with findings of a higher incidence of PPROM development after FLP and a lower proportion of cases of stage I TTTS in the group in which FLP was implemented at an early GA, may partially explain the association between FLP being implemented at an earlier GA and the poorer perinatal outcomes in our case series. When TTTS received amnioreduction only, the GA at diagnosis also was found to be a significant factor for infant survival at 4 weeks of age [27]. So, an early GA at diagnosis was a significant risk factor for lower fetal survival in TTTS if either FLP or amnioreduction was chosen as the treatment method.

A report revealed that individuals with TTTS who received FLP at a GA younger than 17 weeks had similar outcomes to those with TTTS who received FLP at a GA of 17 to 26 weeks. However, the report did not include individuals with stage I TTTS in the group who received FLP at a GA younger than 17 weeks [28]. Another study discovered that FLP performed at a GA of earlier than 18 weeks in individuals with TTTS at Quintero stages II–IV and stage I with associated symptoms was not associated with an increased rate of very preterm delivery and PPROM development after FLP or with lower neonatal survival [14]. The results of the aforementioned studies differed from those of another recent report [7]. These differences in the reported effect of GA at the time of FLP for TTTS on the outcomes of FLP are likely ascribed to the fact that the relevant studies investigated different variables of included cases (e.g., selective growth restriction, placental location, size of the trocar, cervical length, and severity of TTTS) [14].

In our earlier series that enrolled 46 individuals with TTTS after FLP, we discovered that an earlier GA at delivery was the most significant predictor of neonatal neurodevelopment disability [29]. Other studies have reported that an earlier GA at birth was associated with poorer neurological outcomes in cases involving TTTS with FLP [30,31]. FLP being implemented at an advanced GA was also reported to be a risk factor for neurodevelopment impairment in TTTS after FLP [30]. However, the authors indicated that the discovered association required further clarification and was only speculative [31]. In the present study, the GA at which FLP is performed was not significantly associated with neonatal brain sonographic image anomalies; only GA at delivery (degree of prematurity) was significantly associated with such anomalies.

The present study has the following strengths: First, this study included patients with stages I to IV TTTS diagnosed before a GA of 26 weeks who underwent FLP. Such patients were included because the inclusion of some TTTS cases presenting with Quintero stage I in a randomized clinical trial was previously reported to make the management of those cases heterogeneous [7]. Second, in the present study, all surgeries were performed by a single surgeon (YLC) at a single center, and similar equipment and techniques were used. This likely decreased the heterogeneity of the included cases. The present study also had some limitations. First, the number of cases was small, and the study was retrospective in nature. Second, because our center routinely performs FLP for individuals with stage I TTTS, the number of cases that may regress or progress in severity cannot be estimated.

## 5. Conclusions

By analyzing a consecutive series of cases of Quintero stage I to IV TTTS diagnosed before a GA of 26 weeks and in which FLP was implemented, we discovered that FLP implementation at an earlier GA is a risk factor for lower fetal survival and a higher rate of PPROM development after FLP. In addition, the incidence of neonatal brain ultrasound image anomalies is associated with the degree of prematurity but not with the GA at FLP in individuals with TTTS. Although FLP being implemented at an earlier GA is a risk factor for poorer fetal outcomes in individuals with TTTS, individuals with stage II to IV TTTS should receive FLP without delay. Delaying FLP for cases involving stage I TTTS diagnosed at an early GA without risk factors, such as a short cervical length, may be considered, but whether delaying FLP would improve surgical outcomes and, if so, how long the delay should be may need further trials to answer.

## Figures and Tables

**Figure 1 jcm-12-01900-f001:**
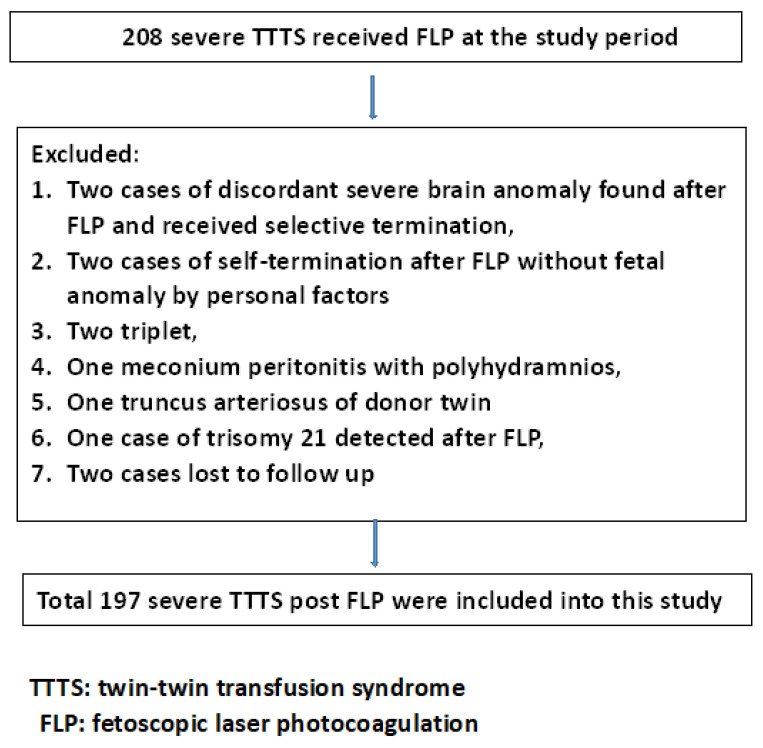
Flowchart of the inclusion and exclusion of patients with severe TTTS who received FLP.

**Table 1 jcm-12-01900-t001:** Basic characteristics of patients with twin–twin transfusion syndrome after fetoscopic laser photocoagulation.

TTTS Receipted FLP (N = 197)		
Maternal age at operation (years old)		31.9 ± 4.7
Gestational age at operation (weeks)		20.6 ± 2.5
Gestational age at delivery (weeks)		31. 5 ± 5.5
Interval from operation to delivery (days)		76.3 ± 41.1
Quintero stage (number)		
	I	30 (15.2%)
	II	68 (34.5%)
	III	69 (35.0%)
	IV	30 (15.2%)
Mean cervical length before FLP (cm)		3.2 ± 0.9
Cerclage after FLP (numbers)		28 (16.6%)
MVP of amniotic fluid of recipient twin (cm)		11.6 ± 3.0
Mainly anterior placenta		57 (28.9%)
Two survivors (numbers)		117 (59.4%)
At least one survivor (numbers)		164 (83.2%)
No survival (numbers)		33 (16.8%)
PPROM within 21 days after FLP (numbers)		27 (13.7%)
Neonatal brain sonographic anomaly (numbers)		17/285 (5.96%)
Severe neonatal brain sonographic anomaly (numbers)		7/285 (2.46%)

TTTS: twin–twin transfusion syndrome; FLP: fetoscopic laser photocoagulation; MVP: maximum vertical pocket; PPROM: preterm premature rupture of membranes.

**Table 2 jcm-12-01900-t002:** Characteristics of patients with twin–twin transfusion syndrome with an early gestational age (≤20 weeks) and late gestational age (>20 weeks) at the time of fetoscopic laser therapy.

	TTTS with Early GA at FLP (N = 81)	TTTS with Late GA at FLP (N = 116)	*p*
Maternal age at operation (years old)	31.4 ± 4.3	32.3 ± 4.9	0.34 ^#^
Gestational age at operation (weeks)	18.2 ± 1.2	22.4 ± 1.7	<0.001 ^&^
Gestational age at delivery (weeks)	30.8 ± 6.5	32.0 ± 4.9	0.46 ^#^
Mainly anterior placenta	23 (28.4%)	34(29.3%)	1.0 *
Cervical length	3.3 ± 0.9	3.2 ± 1.0	0.53 ^&^
Cervical cerclage after FLP	7(8.6%)	21 (18.1%)	0.066 *
MVP of recipient twin	10.1 ± 2.3	12.6 ± 2.9	<0.001 ^&^
Stage I TTTS	6 (7.4%)	24 (20.7%)	0.015 *
Stage II TTTS	30 (37.0%)	38 (32.8%)	0.546 *
High Quintero stage (III or IV)	45 (55.6%)	54 (46.6%)	0.25 *
At least one survivor	61 (75.3%)	103 (88.8%)	0.019 *
Two survivors	41 (50.6%)	76 (65.5%)	0.040 *
Donor survival (numbers)	49 (60.5%)	85 (73.3%)	0.064 *
Recipient survival (numbers)	55 (67.9%)	96 (82.8%)	0.017 *
PPROMs within 21 days after FLP	16 (19.8%)	10 (8.6%)	0.032 *
Interval from operation to delivery (days)	89.3 ± 46.4	67.3 ± 34.4	<0.001 ^#^
Neonatal brain sonographic anomaly	4 (4.6%)	13 (7.1%)	0.20 *
severe neonatal brain sonographic anomaly	1/131(1.2%)	6/154 (5.2%)	0.24 *

TTTS: twin–twin transfusion syndrome; GA: gestational age; FLP: fetoscopic laser photocoagulation; PPROM: preterm premature rupture of membranes. ^#^: Mann–Whitney *U* Test; ^&^: Student’s *t*-test; *: Chi-squared rest or Fisher’s exact test.

**Table 3 jcm-12-01900-t003:** Outcomes of TTTS after FLP among different Quintero stages.

Quintero Stage	Stage I (N = 30)	Stage II (N = 68)	Stage III (N = 69)	Stage IV (N = 30)	*p*
At least one survivor	28 (93.3%)	58 (85.3%)	55 (79.7%)	23 (76.7%)	0.268 *
Two survivors	23 (76.7%)	51 (75.0%)	27 (39.1%)	16 (53.3%)	<0.001 *
PPROM after FLP	3 (10.0%)	8 (11.8%)	9 (13.0%)	6 (20.0%)	0.659 *
GA at delivery (weeks)	32.4 ± 4.8	32.0 ± 5.4	30.5± 5.6	31.8 ± 6.2	0.272 ^#^

TTTS: twin–twin transfusion syndrome; GA: gestational age; FLP: fetoscopic laser photocoagulation; PPROM: preterm premature rupture of membranes. *: Chi-squared test; ^#^: one-way ANOVA.

**Table 4 jcm-12-01900-t004:** Antenatal variables for preterm premature rupture of membranes within 21 days of FLP in severe cases of TTTS.

Variable	Odd Ratio (95% CI)	*p*	Adjusted Odds Ratio (95% CI)	*p*
GA at FLP (weeks)	0.80 (0.64~0.99)	0.043	0.82 (0.68~0.99)	0.035
MVP of recipient (cm)	NS	0.76	NS	NS
Mainly anterior placenta	NS	0.14	NS	NS
Cervical length (cm)	0.63 (0.40~0.99)	0.026	0.59 (0.37~0.93)	0.023

FLP: fetoscopic laser photocoagulation; TTTS: twin–twin transfusion syndrome; MVP: maximum vertical pocket. NS: not significant (*p* > 0.05).

**Table 5 jcm-12-01900-t005:** Antenatal variables for the survival of both twins after FLP in severe cases of TTTS.

Variable	Odds Ratio (95% CI)	*p*	Adjusted Odds Ratio (95% CI)	*p*
GA at FLP (weeks)	1.18 (1.05~1.33)	0.006	1.18 (1.03~1.34)	0.013
MVP of recipient (cm)	NS	0.059	NS	NS
Mainly anterior placenta	NS	0.34	NS	NS
Cervical length (cm)	1.65(1.16~2.34)	0.006	1.62 (1.14~2.31)	0.007
Stage I TTTS	2.55 (1.04~6.27)	0.041	NS	NS
Stage II TTTS	2.86 (1.50~5.48)	0.001	NS	NS
Stage III TTTS	0.27 (0.15~0.50)	<0.001	0.29 (0.16~0.56)	<0.001
Stage IV TTTS	NS	0.47	NS	NS

FLP: fetoscopic laser photocoagulation; TTTS: twin–twin transfusion syndrome; GA: gestational age; MVP: maximum vertical pocket. NS: not significant (*p* > 0.05).

**Table 6 jcm-12-01900-t006:** Antenatal variables for the survival of one twin after FLP in severe cases of TTTS.

Variable	Odds Ratio (95% CI)	*p*	Adjusted Odds Ratio (95% CI)	*p*
GA at FLP (weeks)	1.28 (1.03~1.57)	0.023	1.31 (1.10~1.57)	0.003
MVP of recipient (cm)	NS	0.75	NS	NS
Mainly anterior placenta	NS	0.55	NS	NS
Cervical length (cm)	2.22 (1.40~3.54)	0.001	2.27 (1.45~3.57)	<0.001

FLP: fetoscopic laser photocoagulation; TTTS: twin–twin transfusion syndrome; GA: gestational age; MVP: maximum vertical pocket. NS: not significant because *p* > 0.05.

**Table 7 jcm-12-01900-t007:** Variables for neonatal brain sonographic anomalies in patients with TTTS treated using FLP.

Variable	Odd Ratio (95% CI)	*p*	Adjusted Odds Ratio (95% CI)	*p*
GA at FLP (weeks)	NS	0.142	NS	NS
MVP of recipient (cm)	NS	0.71	NS	NS
Mainly anterior placenta	NS	0.43	NS	NS
GA at delivery (weeks)	0.86 (0.78~0.96)	0.008	0.90 (0.82~0.98)	0.016

FLP: fetoscopic laser photocoagulation; TTTS: twin–twin transfusion syndrome; GA: gestational age; MVP: maximum vertical pocket. NS: not significant (*p* > 0.05).

## Data Availability

Data can be obtained from the corresponding author upon reasonable request.

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
