# Peer review of "Effect of Gestational Age at Fetoscopic Laser Photocoagulation on Perinatal Outcomes for Patients with Twin–Twin Transfusion Syndrome"

_jcm, 2023, doi:10.3390/jcm12051900_

Round 1
Reviewer 1 Report
In this study, the authors aimed to evaluate the gestational age at fetoscopic laser photocoagulation effect on the perinatal outcome. To do so the authors retrospectively reviewed all TTTS cases treated with laser at their center between 2007-2022. The primary outcomes were defined as: premature rupture of membranes (PROMs) within 21 days after laser, survival after 28 days after delivery, gestational age at delivery and neonatal brain sonographic image findings within one month after delivery. Following the study, the authors concluded that late photocoagulation, performed after 20+6 days is better with decreased rates of fetal mortality and PROMs.
My concerns:
I don’t understand the logic of evaluating the gestational age at diagnosis of TTTS cases as this is a fixed variable that can not be changed by us. The real question for stage 1-2 TTTS is if postponing the laser improves the outcome (assuming we wonlt delay treatment for stage 3-4). Also, the study requires an extensive English review.
Abstract - Results should follow the aim of the study. Only 15% of the cohort had stage 1 TTTS – they are the only group presented. Also – no abbreviation of MVP
Introduction- the data is presented however extensive English review is needed
Methods – for all cases, and especially for stage 1 TTTS, what was the time interval from diagnosis to treatment? Although this data is presented in table 2, it should be evaluated for every stage separately and used as a covariate for outcome results.
Results – the numbers in the flowchart of the study cohort do not add up.
Results should be evaluated separately for every TTTS stage or at least entered in the logistic regression.
Severe TTTS is defined as stage 3-4 TTTS. How is stage 1 a covariate? (table 4)
Author Response
In this study, the authors aimed to evaluate the gestational age at fetoscopic laser photocoagulation effect on the perinatal outcome. To do so the authors retrospectively reviewed all TTTS cases treated with laser at their center between 2007-2022. The primary outcomes were defined as: premature rupture of membranes (PROMs) within 21 days after laser, survival after 28 days after delivery, gestational age at delivery and neonatal brain sonographic image findings within one month after delivery. Following the study, the authors concluded that late photocoagulation, performed after 20+6 days is better with decreased rates of fetal mortality and PROMs.
My concerns:
I don’t understand the logic of evaluating the gestational age at diagnosis of TTTS cases as this is a fixed variable that can not be changed by us.
Answer: thanks for commits: because some reports about the TTTS received FLP using the GA at diagnosis as a variable to evaluate the outcomes, as we know the timing of occurrence of TTTS may be hard to define in some cases. In our center, we usually performed FLP within 1 to 2 days after confirmation of the diagnosis. So the GA at diagnosis may be very similar to the GA at FLP. And in some of TTTS, FLP would not be chosen by parents due to variable factors, GA of diagnosis may have value for prognosis.
The real question for stage 1-2 TTTS is if postponing the laser improves the outcome (assuming we wonlt delay treatment for stage 3-4). Also, the study requires an extensive English review.
Answer: We have our manuscript edited by a native English speaker, thanks.
Abstract - Results should follow the aim of the study. Only 15% of the cohort had stage 1 TTTS – they are the only group presented. Also – no abbreviation of MVP
Answer: tanks, we make a change of MVP as “maximum vertical pocket” without aberration and refine the abstract.
Introduction- the data is presented however extensive English review is needed
Answer: We have our manuscript edited by a native English speaker, thanks.
Methods – for all cases, and especially for stage 1 TTTS, what was the time interval from diagnosis to treatment? Although this data is presented in table 2, it should be evaluated for every stage separately and used as a covariate for outcome results.
Answer: As we described previously, the timing of occurrence of TTTS may be hard to define in some cases. We usually performed FLP within 1 to 2 days after confirmed diagnosis; though some cases may be diagnosed at the referral hospital, but due to Taiwan is a traffic convenience island patients always come to our hospital within 1 or 2 days after referral and we provide the service 24 hours a day and seven days a week. So the GA at FLP was very close to the GA of diagnosis.
Results – the numbers in the flowchart of the study cohort do not add up.
Answer: thanks, we have corrected it.
Results should be evaluated separately for every TTTS stage or at least entered in the logistic regression.
Answer: we have added a table 3 for the outcomes according to Quintero stage.
And we found different Quintero stage would be a significant variant for two fetal survival rate after FLP, so we add Quintero stage to the logistic model to predict two fetal survival rate in the revised manuscript.
Severe TTTS is defined as stage 3-4 TTTS. How is stage 1 a covariate? (table 4)
Answer: thanks, we have deleted stage I TTTS as a covariable in logistic regression models.
Reviewer 2 Report
The article “Effect of gestational age at fetoscopic laser photocoagulation on perinatal outcomes for twin-to-twin transfusion syndrome” is an interesting paper, however due to poor English sometimes difficult to understand – it requires extensive language editing. Such mistakes as “a PROMs”, “to categorical the severity” “TTTS receipted FLP” and many others are simply unacceptable.
As for the scientific merit:
- The group o FLP procedures is quite impressive (197 TTTS treated cases) and the observation time as well (2005-2022), therefore it is worth to improve the paper and have it published
- Since the analyzed period includes 17 years the authors should briefly mention how many people perform the FLP procedures in the hospital, because the learning curve of FLP is a well known factor influencing the results. Additionally, over the long period – where there any changes in qualification?
- Material and methods section: requires improvement – there are no inclusion/exclusion criteria mentioned – instead the authors are presenting the excluded cases in results section (while those excluded should never be in the study in the first place – for example fetusus with trisomy 21 or major anomalies or cord occlusion cases)
- Why was Solomon technique applied for some and for some not? Any changes over time? Any variabilities between operators? The authors also do not compare the efficacy of both techniques – even if there was no difference it should be mentioned
- Table 1 incorporates valuable information, however Table 2 for example does not give any information on TTTS stage II (only I and III+IV) – why?
- The authors divided early/late FLP according to the average GA at FLP – why? It should rather be divided as <20weeks and>20weeks. In addition it is well known that most difficult cases with poor outcomes are those performed prior to 18 weeks – were there any such cases in your material? Should be listed and analyzed separately. GA at FLP in early/late is very similar (19.3 vs 20.7; anyways isn’t 20.7=21 weeks? How is that calculated?) due to the suggested cut-off point 20,6weeks – I would suggest recalculating your data
- Cases with cervical cerclage should be analyzed separately – we only know that cervical length was similar in both early and late FLP and the number of patients with cerclage in each group is given – no further information
- Why was the FLP procedure performed in TTTS stage I (almost 9% of cases)? any changes over time? It especially refers to early FLPs – it wasn’t necessary, especially since we know the results of early FLP are worse when perinatal outcome is compared
- I do not understand “X” in tables 3, 4, 5 an 6 – next to no data the authors put insignificant p values – where does that come from? The construction of such tables is questionable
- There is no information on amnioreduction at the time of FLP
- The change over the years should be described, as 17 years in clinical practice is a long time and procedures change
- Can the authors incorporate the procedure protocol in the material and methods section? Are any NSAIDs (such as indomethacin) applied at the time of procedure? Any antibiotics?
- Page 5 lines 141-143: are those copy/paste mistakes? They seem not to be the part of the study
- The conclusions are controversial: the authors suggest that FLP should not be delayed and should be performed prior to 26 weeks – the recommendations say the procedure SHOULD BE performed prior to 26 weeks, as it is less effective later on and more difficult. It is not the conclusion of this particular study but an international consensus. What should be discussed here is in whom we might delay the procedure in order not to perform early FLPs with poorer outcome – the authors never discussed and never presented data showing FLP was dome >26 weeks of gestation – should be rewritten.
To summarize – it could be a valuable article if thoroughly corrected and improved.
Author Response
The article “Effect of gestational age at fetoscopic laser photocoagulation on perinatal outcomes for twin-to-twin transfusion syndrome” is an interesting paper, however due to poor English sometimes difficult to understand – it requires extensive language editing. Such mistakes as “a PROMs”, “to categorical the severity” “TTTS receipted FLP” and many others are simply unacceptable.
As for the scientific merit:
- The group o FLP procedures is quite impressive (197 TTTS treated cases) and the observation time as well (2005-2022), therefore it is worth to improve the paper and have it published
- Since the analyzed period includes 17 years the authors should briefly mention how many people perform the FLP procedures in the hospital, because the learning curve of FLP is a well known factor influencing the results. Additionally, over the long period – where there any changes in qualification?
Answer: There was only one surgeon (YL Chang) to perform the FLP for TTTS in our center. From October 2007 to 2015 March, we perform FLP by SLPCV only; after that we performed FLP by SLPCV with Solomon technique in some cases with suspicious anastomoses, especially with anterior placenta.
- Material and methods section: requires improvement – there are no inclusion/exclusion criteria mentioned – instead the authors are presenting the excluded cases in results section (while those excluded should never be in the study in the first place – for example fetusus with trisomy 21 or major anomalies or cord occlusion cases)
Answer: we have added the inclusion and exclusion criteria in the material and methods section.
- Why was Solomon technique applied for some and for some not? Any changes over time? Any variabilities between operators? The authors also do not compare the efficacy of both techniques – even if there was no difference it should be mentioned.
Answer: From October 2005 to 2015 March, we perform FLP by SLPCV only; after that we performed FLP by SLPCV with Solomon technique in some cases with suspicious anastomoses especially with anterior placenta. After 2015, there were a total 99 TTTS included in this study: there were no significant differences in the outcomes as one and two fetal survivals, PROMS after FLP, and GA at delivery. We add those data in the supplementary file.
- Table 1 incorporates valuable information, however Table 2 for example does not give any information on TTTS stage II (only I and III+IV) – why?
Answer: we have added the information of stage II TTTS in table 2.
- The authors divided early/late FLP according to the average GA at FLP – why? It should rather be divided as <20 weeks and>20weeks. In addition it is well known that the most difficult cases with poor outcomes are those performed prior to 18 weeks – were there any such cases in your material? Should be listed and analyzed separately. GA at FLP in early/late is very similar (19.3 vs 20.7; anyways isn’t 20.7=21 weeks? How is that calculated?) due to the suggested cut-off point 20,6 weeks – I would suggest recalculating your data
Answer: thanks for the comments: we have re-calculated the data by a cutoff point at GA 20 weeks in the revised manuscript.
- Cases with cervical cerclage should be analyzed separately – we only know that cervical length was similar in both early and late FLP and the number of patients with cerclage in each group is given – no further information
Answer: 1. We have added the number of TTTS with cerclage in table 1.
- Because we perform cerclage in cases of TTTS with short cervical length, and short cervical length is a risk factor for less fetal survival, more risk of PPROMs after FLP, and smaller GA at delivery. The results have been added in the discussion section. Thanks.
- Why was the FLP procedure performed in TTTS stage I (almost 9% of cases)? any changes over time? It especially refers to early FLPs – it wasn’t necessary, especially since we know the results of early FLP are worse when perinatal outcome is compared.
Answer: We have not yet changed our protocol due to the criteria formed at 2005, after the analysis our data we may try to make a revision of our protocol.
- I do not understand “X” in tables 3, 4, 5 an 6 – next to no data the authors put insignificant p values – where does that come from? The construction of such tables is questionable
Answer: Thank for comments: X mean the Odd ratio do not present due to p value was not significant. In order to reduce confusion, we delete X in in tables 3, 4, 5 an 6 and change to NS: NS: “not significant” due to p value > 0.05.
- There is no information on amnioreduction at the time of FLP
Ans: We have added “amnioreduction was performed to reduce the amount of amniotic fluid to the MVP of the recipient twin was less than 8 cm.” in the material and method section.
- The change over the years should be described, as 17 years in clinical practice is a long time and procedures change
Answer: From October 2005 to 2015 March, we perform FLP by SLPCV only; after that, we performed FLP by SLPCV with Solomon technique in some cases with suspicious anastomoses, especially with anterior placenta.
- Can the authors incorporate the procedure protocol in the material and methods section? Are any NSAIDs (such as indomethacin) applied at the time of procedure? Any antibiotics?
Answer: We have added the “surgical procedure.” In the material and method section.
- Page 5 lines 141-143: are those copy/paste mistakes? They seem not to be the part of the study
Answer: thanks, there was a mistake and we have corrected it in the revised manuscript.
- The conclusions are controversial: the authors suggest that FLP should not be delayed and should be performed prior to 26 weeks – the recommendations say the procedure SHOULD BE performed prior to 26 weeks, as it is less effective later on and more difficult. It is not the conclusion of this particular study but an international consensus. What should be discussed here is in whom we might delay the procedure in order not to perform early FLPs with poorer outcome – the authors never discussed and never presented data showing FLP was dome >26 weeks of gestation – should be rewritten.
Answer: Thanks, we have corrected it.
To summarize – it could be a valuable article if thoroughly corrected and improved.
Reviewer 3 Report
At first glance the manuscript appears interesting; however at a more detailed scrutiny several issues appear. Some of them are: a) It is difficult to follow; b) it is not clear; c) the numbers and data do not match in different sections of the manuscript (In the abstract the time of the study ranges from October 2007 to September 2022; in Material and Methods it is from October 2005 and September 2022. The total number of cases studied following those excluded is 197 in the abstract, 207 (228 – 11 = 217) and 197 in the Results); d) there are many grammar, syntax, and spelling errors; e) the rationale of the 20.6 weeks is not clear; f) The Discussion section is disorganized.
My suggestions are:
Please check the numbers and data.
It is PPROM and not PROM
Clarify the reason(s) of delivery
Why did you select 20.6 weeks?
Avoid the redundancies between the text and the tables
Please consider rewriting the manuscript and do the analysis with the entire group before and after X weeks. (Clarify why you select 20.6 weeks to divide the two groups). Next you can divide the data by stage.
Author Response
At first glance the manuscript appears interesting; however at a more detailed scrutiny several issues appear. Some of them are: a) It is difficult to follow; b) it is not clear; c) the numbers and data do not match in different sections of the manuscript (In the abstract the time of the study ranges from October 2007 to September 2022; in Material and Methods it is from October 2005 and September 2022. The total number of cases studied following those excluded is 197 in the abstract, 207 (228 – 11 = 217) and 197 in the Results); d) there are many grammar, syntax, and spelling errors; e) the rationale of the 20.6 weeks is not clear; f) The Discussion section is disorganized.
My suggestions are:
Please check the numbers and data.
Answer: We have corrected the including number in figure 1, the study period in the abstract section, and our manuscript be edited by a native English speaker. We also use gestational age as 20 weeks to divide cases as early and late gestational age to analyze the result in the revised manuscript. Many thanks.
It is PPROM and not PROM
Answer: we have changed PROMs to PPROMs in the revised manuscript.
Clarify the reason(s) of delivery
Answer: If no preterm labor delivery: we delivered TTTS post SLP no more than GA 36 weeks in two survival cases if parents agree. In single survival after FLP, we delivered the babies no more than GA 40 weeks.
Why did you select 20.6 weeks?
Answer: We use gestational age as 20 weeks to divide cases as early and late gestational age to analysis the result in the revised manuscript
Avoid the redundancies between the text and the tables
Answer: we have corrected it.
Please consider rewriting the manuscript and do the analysis with the entire group before and after X weeks. (Clarify why you select 20.6 weeks to divide the two groups). Next you can divide the data by stage.
Answer: We use gestational age as 20 weeks to divide cases as early and late gestational age to analyze the result in the revised manuscript. And we also analyze the results by Quintero stage in the revised manuscript.
Round 2
Reviewer 1 Report
I agree that the manuscript is greatly improved.
I only ask the authors to rephrase their conclusion to delay FLP in cases with early stage 1 TTTS. This can not be concluded from this study
Author Response
I agree that the manuscript is greatly improved.
I only ask the authors to rephrase their conclusion to delay FLP in cases with early stage 1 TTTS. This can not be concluded from this study
Answer: Thanks, we have revised the conclusion according to the suggestion.
Reviewer 3 Report
Dear Authors,
The manuscript is greatly improved. I congratulate you!
My suggestions are:
Please clarify how many stage I TTTS following 20 weeks' gestation had the largest vertical pocket in the recipient less than 10 cm. The reason is that Quintero staging was developed in 1998. This stage uses 8 cm as criteria of polyhydramnios at any gestational age. One of the largest studies reported in 2001 (TTTS Registry published in the Am J Obstet Gynecol) used a value of 10 cm or more following 20 weeks' gestation for the definition of TTTS. The same criteria were reported in the paper by Senat et al in 2004. Most of the centers today use the same criteria. Please add the amnioreduction study in the references (Am J Obstet Gynecol 2001; 185:708-15).
We are now aware that monochorionic twins after 20 weeks' gestation with the largest vertical pocket less than 10 cm, do not need any form of therapy and they are not considered clearly TTTS.
My suggestion is that you add this concept in the Discussion section. Even if you have treated with Laser some of your patients with the largest MVP < 10cm it would be extremely important to report it.
This concept will be novel and one of the most important concepts of your manuscript. I believe that it will get a lot of attention and your manuscript will be highly cited as the 1st one that has emphasized the concept.
Please review one more time the Discussion section because it appears slightly disorganized.
Author Response
The manuscript is greatly improved. I congratulate you!
My suggestions are:
Please clarify how many stage I TTTS following 20 weeks' gestation had the largest vertical pocket in the recipient less than 10 cm. The reason is that Quintero staging was developed in 1998. This stage uses 8 cm as criteria of polyhydramnios at any gestational age. One of the largest studies reported in 2001 (TTTS Registry published in the Am J Obstet Gynecol) used a value of 10 cm or more following 20 weeks' gestation for the definition of TTTS. The same criteria were reported in the paper by Senat et al in 2004. Most of the centers today use the same criteria. Please add the amnioreduction study in the references (Am J Obstet Gynecol 2001; 185:708-15).
Answer: Thanks for the comment, the study (Am J Obstet Gynecol 2001; 185:708-15) found the GA at diagnosis was a significant factor to predict fetal survival rates. So we add this reference to this study.
We are now aware that monochorionic twins after 20 weeks' gestation with the largest vertical pocket less than 10 cm, do not need any form of therapy and they are not considered clearly TTTS.
My suggestion is that you add this concept in the Discussion section. Even if you have treated with Laser some of your patients with the largest MVP < 10 cm it would be extremely important to report it.
This concept will be novel and one of the most important concepts of your manuscript. I believe that it will get a lot of attention and your manuscript will be highly cited as the 1st one that has emphasized the concept.
Answer: Thanks for the comments. We included 30 cases of stage I TTTS in this study and 24 cases in which the GA at FLP was more than 20 weeks. After dividing cases with recipient MVP less and more than 10 cm: we found there was no difference in fetal survival rates, the incidence of PPROMs, and GA at delivery between the two groups. The cases of stage I TTTS with poorer prognosis after FLP were found in the groups with FLP at GA less than 20 weeks. So by our data, TTTS stage I cases received FLP after GA 20 weeks between recipient twins with MVP more and less than 10 cm with comparable outcomes.
TTTS stage I received FLP performed >20 weeks
|
|
MVP of recipient ≤ 10 cm (N=7) |
MVP of recipient >10 cm (N=17) |
p |
|
PPROMs |
0 |
0 |
NA |
|
Two survival |
6 (85.7%) |
14 (82.4%) |
1.0 |
|
At least one survival |
7 (100%) |
17 (100%) |
NA |
|
GA at delivery (weeks) |
34.8 |
32.9 |
0.26 |
Please review one more time the Discussion section because it appears slightly disorganized.
Answer: Thanks to the reviewer, we have revised our discussion section to let it more organized.